# QoS-Based Service-Time Scheduling in the IoT-Edge Cloud

**DOI:** 10.3390/s21175797

**Published:** 2021-08-28

**Authors:** Briytone Mutichiro, Minh-Ngoc Tran, Young-Han Kim

**Affiliations:** School of Electronic Engineering, Soongsil University, Seoul 06978, Korea; briyt.mutichiro@dcn.ssu.ac.kr (B.M.); mipearlska1307@dcn.ssu.ac.kr (M.-N.T.)

**Keywords:** IoT-edge cloud, resource scheduling, quality of service (QoS), ant colony optimization (ACO)

## Abstract

In edge computing, scheduling heterogeneous workloads with diverse resource requirements is challenging. Besides limited resources, the servers may be overwhelmed with computational tasks, resulting in lengthy task queues and congestion occasioned by unusual network traffic patterns. Additionally, Internet of Things (IoT)/Edge applications have different characteristics coupled with performance requirements, which become determinants if most edge applications can both satisfy deadlines and each user’s QoS requirements. This study aims to address these restrictions by proposing a mechanism that improves the cluster resource utilization and Quality of Service (QoS) in an edge cloud cluster in terms of service time. Containerization can provide a way to improve the performance of the IoT-Edge cloud by factoring in task dependencies and heterogeneous application resource demands. In this paper, we propose STaSA, a service time aware scheduler for the edge environment. The algorithm automatically assigns requests onto different processing nodes and then schedules their execution under real-time constraints, thus minimizing the number of QoS violations. The effectiveness of our scheduling model is demonstrated through implementation on KubeEdge, a container orchestration platform based on Kubernetes. Experimental results show significantly fewer violations in QoS during scheduling and improved performance compared to the state of the art.

## 1. Introduction

Edge computing offers an extension to the central cloud, bringing cloud capabilities close to the end users for edge application providers to deploy their services. In comparison with cloud computing, the advantages include lower data transmission latency, better bandwidth utilization, and improved privacy of user data in the Edge-Cloud collaboration. Previous research [1,2] has shown that the higher the proximity of the application or service to the user, the better the Quality of Service (QoS) attainable by the user. A major limitation in edge computing vis-à-vis traditional clouds is the limitation in resources. However, the traditional cloud model suffers diminished network performance and inefficiency occasioned by the processing and analysis of big data generated from remote edge devices. This design results in increased pressure on the network created by the traffic pattern leading to poor user QoS. Similarly, in the traditional cloud, services are usually implemented with the same level of availability with less regard for service/application-specific characteristics. In contrast, different Internet of Things (IoT)/Edge applications have various characteristics and performance requirements [3,4,5]. Examples of these edge applications include data analytics, augmented/virtual reality (AR/VR), autonomous driving, smart manufacturing, etc.

Task scheduling in cloud computing works based on the current information of tasks and resources in accordance with a certain strategy in order to establish an appropriate mapping relationship of tasks to appropriate resources. From the perspective of edge computing, the impact of scheduling heterogeneous workloads with diverse resource requirements is challenging. Consider a scenario where multiple devices can connect to an edge server simultaneously within a given period. Consequently, the server may be overwhelmed with computational tasks, resulting in lengthy task queues. This results in congestion, occasioned by increased completion time for all queued jobs, even to the point where the processing delay of jobs at the edge server exceeds that at the edge devices. Furthermore, latency-critical jobs need to be scheduled as soon as they are submitted to avoid any queuing delays, while for best-effort latency-tolerant jobs, they should be allowed to occupy the node cluster when there are idle resources in order to improve cluster utilization [6,7]. These implications become determinants if most edge applications can both satisfy deadlines and each user’s QoS requirements [5].

Exploiting containerized technology to host applications can provide a way to improve the performance of edge computing platforms [8]. Containers are a lightweight application virtualization technology that provides a logical packing mechanism for application abstraction that packages software and dependencies together. In addition to providing a virtual runtime environment based on a single operating system (OS) kernel, containers also support resource sharing across multiple users and tasks concurrently rather than booting an entire OS for each application [9]. This permits agile application deployment, orchestration environment consistency, OS portability, application-centric management, and resource isolation for container-based applications. A high-level view of a containerization framework with a custom scheduler [10] is displayed in Figure 1, which is similar to our proposal from a deployment perspective. Multiple leading cloud computing providers have also adopted containers for the deployment of services directly on their edge platforms: Google Cloud IoT, Microsoft Azure IoT Edge, and Amazon AWS IoT Greengrass [11,12]. Therefore, in this work, we enhance our work in [5] by implementing a real test bed through adopting the KubeEdge orchestration platform that is based on Kubernetes. The summary of our contributions are as follows:We propose a multi-objective model for pod scheduling. The model considers constraints in terms of resource capacity (CPU and memory) and optimizes the processing time overhead, the scheduling cost on nodes, and the number of QoS violations in terms of overall service time for instantiation and scheduling.An enhanced ant colony optimization (ACO)-inspired algorithm (STaSA) is proposed. The algorithm combines multi-objective heuristic information (node utilization, service time, and scheduling cost) based on a pheromone model to improve the request scheduling probability for optimal placement.We design and implement a real-time test bed for our custom STaSA scheduler. The effectiveness of our scheduling model is demonstrated through a deployment on the KubeEdge orchestration platform, and we provide an analysis of the experimental results.

The rest of this paper is organized as follows. In Section 2, we provide an overview of related research works. In Section 3, we formulate our problem, and Section 4 presents an analysis of our model and algorithm solution. Section 5 involves analysis and discussion of the results. This work is concluded in Section 6 by indicating the open issues for us to build a desired collaborative scheduling system for edge computing.

## 2. Related Works

### 2.1. Scheduling Strategies

The compute, storage, task execution status, and network state, among other things, inform the edge scheduling policies. Chen et al. explore a collaborative scheduling mechanism within a device–edge–cloud infrastructure framework [13] based on task characteristics, optimization objectives, and system status. Task splitting and task interdependence are considered to determine local execution, partial offloading, and full offloading for given computing tasks. A pre-emptive fair share cluster scheduler is developed in [6], where the authors leverage containerization to enable pre-emptive and low-latency scheduling in clusters with heterogeneous workloads. Two pre-emption strategies are proposed: immediate and graceful pre-emptions and the effectiveness and tradeoffs evaluated. They investigate that if tasks from short jobs can pre-empt any long tasks, their scheduling can be made simple and fast while long jobs can run on any server in the cluster to maintain high utilization. This approach only supports killed-based task pre-emption, which is not efficient for cluster resource utilization and job performance.

Another study proposed a multi-objective scheduling method [14] that was constrained by users’ budget conditions for online workflow applications. Their strategy both minimizes the execution time of workflows and reduces the budget constraints of users. Ouyang et al. [15] propose a service placement scheduling model for cost-efficient mobile edge computing. Their aim is to minimize service latency and migration costs. The problem is formulated as a stochastic optimization problem that adopts the Lyapunov framework to establish a solution. A weighted bipartite graph matching scheme [16] was used to develop a container scheduling model. In this approach, there must be direct parity between the number of tasks and the number of containers. In a scenario where containers outnumber tasks, the edge orchestrator kills the extra containers; for a converse scenario, hypothetical containers are added to equal the number. With the development of the Internet of Things (IoT), edge computing facilitates largely delay-sensitive and location-aware applications considering the many types of IoT devices. However, the limited resources of edge computing have inspired multiple efforts to enhance efficiency in task execution. Yin et al. [17] study resource scheduling in smart manufacturing, where they propose a container-based task-scheduling model and task-scheduling algorithms with a task-delay constraint. The fog computing system is taken as a hard real-time system, where an acceptable request must be accomplished before the deadline specified by its terminal device. A failure infers that the fog node has inadequate resources for request allocation, and the node should reject the request and inform the terminal device to resubmit the new request within the new deadline. However, the processing capability of an edge node is restricted; thus, small tasks or processing requests with short delay will be prioritized to be processed on the edge infrastructure. This prioritization is realized from the perspective of scheduling conditions of resource allocations for delay-tolerant and delay-sensitive applications [7], which adaptively allocates resources in a mobile cloud computing (MCC) system. Two kinds of resource allocation strategies are adopted coordinately: immediate reservation and advanced reservation to guarantee deadline constraints being satisfied and avoid too much reservation resulting in MCC performance decline. A load-aware resource allocation and task scheduling (LA-RATS) framework is proposed, aiming to significantly reduce the cloudlet’s monetary cost and turnaround time for delay-tolerant applications and increase the deadline satisfaction rate of delay-sensitive.

### 2.2. Kubernetes Related Schedulers

Nguyen et al. [18] present ElasticFog, a mechanism that exploits different scheduling policies in the Kubernetes platform and enables real-time elastic resource provisioning for containerized applications in fog computing. It dynamically assigns resources to each fog node proportionate to network traffic to the application at each location. Consequently, the authors aim to reduce network latency and avoid resource wastage/overprovisioning in areas of low workload demand. Another Kubernetes-based scheduler KEIDS was proposed in [19] for container management with considerations for carbon emissions, interference, and energy consumption. The design aims to achieve effective resource management and job synchronization with minimal interference among co-located containers on the same or different nodes. A scheduling agent approach is explored by [20], where a decentralized Kubernetes-oriented container scheduling model for edge clusters is described. Their approach deploys a scheduling agent on multiple master nodes. Casquero et al. [21] describe a custom scheduler for the Kubernetes orchestrator that distributes the decision logic of the scheduler among edge nodes. Their scheduling agent is supposed to reduce the workload at the control plane of the Kubernetes server. The node filtering and node ranking functions usually executed by the server are undertaken by agents embedded in the edge nodes. A multi-agent scheduling platform receives the node filtering information from all nodes. Then, node ranking is fulfilled through negotiation among the agents in the filtered edge nodes. Considered together, there are several challenges brought in by the edge computing platform [22]: (1) utilization of the computing resources at both the edge cloud and the distant cloud coordinately to fully exploit the system capabilities, and (2) how to perform task scheduling for different classes of applications jointly considering both computing resources and different user QoS requirements.

### 2.3. Scheduling Model

A scheduling policy is responsible for the order of processing requests within the request queue. A common scheduling policy is the First Come First Served (FCFS) approach that is usually adopted as a baseline scheduling strategy [23]. FCFS uses the time sequence of arriving requests to determine the subsequent request to be executed alongside other considerations. A modified version of this approach is presented in [24], where fairness is used as a determinant in the scheduling of task requests. The rationale behind this approach is to prevent over-commitment of resources to requests from a single user creating unfair competition among multiple user requests. Another less frequently used approach is the Earliest Deadline First (EDF) strategy. This strategy prioritizes requests with the smallest remaining deadlines, which are then queued first and allocated resources before other requests. A similar approach is explored in [25] that proposes a task scheduling algorithm with deadline and cost constraints in cloud computing. The authors aim to address a multi-objective optimization problem, i.e., minimize both the total task completion time and cost under deadline constraints. This work considered the Earliest Deadline First scheme [23].

## 3. System Model

This section describes the problem model and the optimization objectives. The modeling of container scheduling through an optimization approach facilitates the problem to be solved mathematically. The parameters of the models and their descriptions are summarized in Table 1. We aim to schedule heterogeneous workloads with diverse resource requirements and QoS constraints. We consider a cluster of *n* nodes with *p* pods. Users request a set of functions *F* with different compute capacity requirements. We do not consider bandwidth requirements as the cluster is locally hosted on a rack server and therefore negates the need to factor internode and inter-container communication considerations. A binary variable $x_{f,i,t}\in \left\{0,1\right\}$ i=1,2,…, p | p ∈P is introduced for the variable for instantiation of function *f* as pod *i* at time *t* as shown mathematically in Equation (1). Another binary variable $y_{ij}$, associating pod *i* to node *j* at time *t*, is also introduced through Equation (2). The sum of all active (running but not scheduled) pods at time *t* is $\varphi^{t}$.
(1)$$\varphi^{t}=\underset{f\in F}{\sum }\underset{i\in P}{\sum }\underset{t\in T}{\sum }x_{f,i,t}$$
(2)$$\overset{p}{\underset{i=1}{\sum }}\overset{n}{\underset{j=1}{\sum }}y_{ij}=\varphi^{t}$$

We consider the following capacity constraints on the scheduling decisions:(3)$$\overset{p}{\underset{i=1}{\sum }}v_{cpu}^{f,t}\cdot y_{ij}\leq r_{cpu}^{j,t}$$
(4)$$\overset{p}{\underset{i=1}{\sum }}v_{mem}^{f,t}\cdot y_{ij}\leq r_{mem}^{j,t}$$

Equations (3) and (4) ensure that the CPU and memory capacity requirements requested by the service pods for allocation do not exceed the total node capacity. The quantities $r_{cpu}^{j,t}$ and $r_{mem}^{j,t}$ represent the total CPU and memory capacities of node *j* at time *t*, respectively. We are able to derive the number of unused resources for a given node *j* using Equations (5) and (6) at time *t*, where ${\hat{r}}_{cpu}^{j,t}$ and ${\hat{r}}_{mem}^{j,t}$ are the available node CPU and memory capacities, respectively.
(5)$${\hat{r}}_{cpu}^{j,t}=r_{cpu}^{j,t}-\left[\overset{p}{\underset{i=1}{\sum }}(y_{ij}\cdot v_{cpu}^{f,t})+\delta_{cpu}\right]$$
(6)$${\hat{r}}_{mem}^{j,t}=r_{mem}^{j,t}-\left[\overset{p}{\underset{i=1}{\sum }}(y_{ij}\cdot v_{mem}^{f,t})+\delta_{mem}\right]$$

We define the utilization $U_{j}^{t}$ of node *j* at time *t* as the weighted relationship between CPU utilization and memory utilization, as indicated by Equation (7), with weight $\varphi_{1}$. Equation (8) indicates the CPU utilization $u^{cpu}$, as a ratio of the available node capacity to the total node capacity and likewise for the memory relationship, $u^{mem}$.
(7)$$U_{j}^{t}=\varphi_{1}\times u^{cpu}+\left(1-\varphi_{1}\right)\times u^{mem}\;\;\;$$
(8)$$where\;u^{cpu}=\frac{{\hat{r}}_{cpu}^{j,t}}{r_{cpu}^{j,t}};\;u^{mem}=\frac{{\hat{r}}_{mem}^{j,t}}{r_{mem}^{j,t}}$$

The cost is an important metric to evaluate the effectiveness of our approach. It is defined in the following Equations (9) and (10). In this formulation, $e_{n}\;$ is a fixed value assigned to the node upon deployment.
(9)$$C_{node}=\frac{1}{1-u^{cpu}}+\frac{1}{1-u^{mem}}$$
(10)$$C_{total}=e_{n}\overset{p}{\underset{i=1}{\sum }}\overset{n}{\underset{j=1}{\sum }}y_{ij}\cdot C_{node}$$

To ensure that the performance of the scheduled jobs is not jeopardized by meeting the respective job deadlines across the allowed time frame, we introduce a value $D_{if}$ that sets a critical deadline for executing a pod *i* of function *f*. Another variable $time_{j}$ is also introduced to represent the processing time of a pod *i* on node *j.* The mathematical representation is denoted below in Equation (11).
(11)$$\overset{p}{\underset{i=1}{\sum }}\overset{n}{\underset{j=1}{\sum }}y_{ij}\times time_{j}\leq \;D_{if}$$

EDF prioritizes requests with the smallest remaining [23] (i.e., earliest) deadlines. The scheduling scheme prioritizes pods with lower remaining time relative to the critical deadline for scheduling. Further, the scheme is biased towards nodes that have a lower processing capacity, i.e., more resources to schedule these pods *p*. The process is conducted incrementally from the perspective of the pods in the scheduling queue until all the node cores are busy. To achieve our objective function *L*, we adopt a linear weighting method. In the function as presented in Equation (12), $\theta_{1}$ is the weight factor of utilization, $\theta_{2}$ is the weight factor of time, and $\theta_{3}$ is the weight factor of the cost.
(12)$$L=\theta_{1}U_{j}^{t}+\theta_{2}time_{j}+\theta_{3}C_{total}$$

## 4. STaSA Algorithm Implementation

The above-defined problem is a typical NP-hard problem, and we address this by means of a heuristic algorithm in order to obtain a near-optimal solution. Papadimitriou et al. [26] describe a Combinatorial Optimization (CO) problem *Z* = (Ω, *g*), as an optimization problem in which is given a finite set of solutions Ω (also called search space) and an objective function *g*: Ω → *R +* that assigns a positive cost value to each of the solutions. The goal is either to find a solution of minimum cost value or a good enough solution in a reasonable amount of time. Ant colony optimization (ACO) is a swarm optimization technique adopted to approximate discrete optimization solutions to hard combinatorial optimization (CO) problems. The approach has been applied to many classical problems such as the TSP, scheduling problems, and recently in cell placement problems and communication networks designs. The inspiring source of ACO algorithms are real ant colonies based on observations of the ants’ foraging behavior. The fundamental characteristic is the indirect communication between the ants via chemical pheromone trails, which enables them to find the shortest path between their nest and food sources. It is a probability-based approach usually adopted for discrete optimization problems, e.g., determining the shortest path in a graph theory [27]. In the search for food, an ant excretes pheromones along a path, which fellow ants use to guide them based on the pheromone concentration. The potency of the pheromone trail is proportional to the quality and quantity of the food the specific ant found, and consequently, with a higher probability, all the remaining colony members will converge along the path of the highest pheromone concentration. This pheromone model consists of a vector of model parameters *T* called pheromone trail parameters. The pheromone trail parameters $\tau_{i}\in T$, which are usually associated with components of solutions, have values $\tau_{i}$, called pheromone values. The pheromone model, based on probability, is used to generate solutions by aggregating them from a finite set of solution components. During execution, ACO algorithms update the pheromone values using previously generated solutions. For a given population of ants and an array of possible paths, each ant determines its path according to the concentration of pheromone trail in each path from the available paths. Generally, the ACO approach has two key points of iteration in the search for a solution: candidate solutions are constructed using a pheromone model, that is, a parametrized probability distribution over the solution space; and the candidate solutions are used to modify the pheromone values in a way that is biased future sampling toward high-quality solutions.

There are several works that use ACO to solve problems for scheduling containers and virtual machines. A variant of ACO is used in [28] to implement schedulers for a software container system. Hafez et al. [29] deploy a modified ACO algorithm with the objective of improving response time and throughput. Other works with either baseline or modified versions of ACO [30,31]. Table 2 is a summary of the state-of-the-art ACO-related research studies as shown below.

## 5. Initializing Pheromones

The objective of this section is to maximize node utilization, minimize the cost, and optimize the service time. This can be determined by conducting a search for optimal pod placement on available nodes. Choosing the node for the next pod can be determined through the pheromone probability relationship $p_{j}^{k}$, shown in Equation (13), which represents the probability that an ant will select the placement of a pod *i* on node *j*; here $\tau_{j}\left(t\right)$ is the node *j* pheromone value at time *t*, $time_{j}$ is the service time of node *j*, and $C_{total}$ is the cost value of node *j*. The exponents *α*, *β*, and *γ* are positive parameters whose values determine the relation between pheromone information and heuristic information ($time_{j}$,$\;C_{total})$, while *k* represents the size of the ant colony, i.e., number of ants.
(13)$$p_{j}^{k}=\frac{{\left[\tau_{j}\left(t\right)\right]}^{\alpha }{\left[time_{j}\right]}^{\beta }{\left[C_{total}\right]}^{\gamma }}{\sum_{N}{\left[\tau_{l}\left(t\right)\right]}^{\alpha }{\left[time_{l}\right]}^{\beta }{\left[C_{total\_l}\right]}^{\gamma }}$$

The initial pheromone value on the path between the pod *i* and the node *j* is calculated as Equation (14), with *t* = 0 taken as the base case for future updates.
(14)$$\tau_{ij}\left(t\right)=U_{j}^{t}:|\;t=0$$

When an ant matches the corresponding node for all pods, an update is performed on the mapping path of the scheduling scheme locally. This is conducted by Equation (15) as follows:(15)$$\tau_{ij}\left(t+1\right)=\left(1-\rho \right)\tau_{ij}\left(t\right)+\Delta \tau_{ij}$$

The amount of pheromone released by an ant due to the quality of the placement is represented as $\Delta \tau_{ij}$ in Equation (16). Here, *A* represents the size of the ant colony, while ρ is the pheromone volatilization factor, indicative of the degree of volatilization per unit time, while (1−ρ) shows the degree of residual pheromone.
(16)$$\Delta \tau_{ij}=\overset{A}{\underset{k=1}{\sum }}\Delta \tau_{ij}^{k}$$

The greater the volatilization factor, the faster the pheromone volatilizing and the smaller the effects of the previous search solution on the present one. The amount of pheromone released by an ant due to the quality of the placement is represented as $\Delta \tau_{ij}\;$ in Equation (16). $\Delta \tau_{ij}^{k}$ is the amount of pheromone left behind during pod-to-node mapping and is related to the objective function as shown in Equation (17), with *Q* as a heuristic constant. *L* represents the fitness of the scheduling of pod *i* on node *j*. If ant *k* went along a path *ij*, it would contribute to the increment of the pheromone on the path. This equation shows that the better solution has a lower evaluation value, and more pheromones on the corresponding path.
(17)$$\Delta \tau_{ij}^{k}=\frac{Q}{L}$$

## 6. Fundamentals of KubeEdge

KubeEdge [32] is an open-source system that provides orchestration and management functions for containerized applications to edge clusters. It is a Cloud Native Computing Foundation (CNCF) sandbox project, designed to extend the Kubernetes ecosystem from cloud to edge. It avails core infrastructure support for networking, application deployment, and metadata synchronization between the cloud and edge. Furthermore, it posits to offer a complete end-to-end edge computing solution anchored on Kubernetes with separate cloud and edge core modules, both of which are open-source. KubeEdge, through the Eclipse Mosquitto message broker, supports the MQTT protocol, which makes it suitable for IoT/resource-constrained device communication. KubeEdge aims to address challenges of large memory footprints occasioned by IoT big data, improve cloud-to-edge network reliability, context-aware offloading, and overall efficiency and scalability. It has multiple components but of interest to our work are two components: Edged and Edge controller. The former is responsible for the pod management, and as an edge node module that manages pod lifecycle, it functions to deploy containerized workloads or applications at the edge node. Those workloads could perform any operation from simple telemetry data manipulation to analytics or ML inference. The Edge controller is the bridge between the Kubernetes API server and edge core. It avails several functions, including both a downstream (K8s API server to edgecore) and an upstream controller. More specifically, these include synchronization of events (node, pod, and configmap), resource status, and subscribe messages. Based on this, we design our scheduler as a plug-in between the K8s Api server and the cloudcore, as shown in Figure 2.

## 7. STaTA Scheduler

Figure 3 shows a high-level architecture of our proposed scheduler. The key characteristic of our design is the implementation of a QoS scheduler that adopts the threshold approach [33] to ensure service scheduling and allocation of resources to pods does not violate the predefined QoS values on cost and execution time based on Equation (11). The workflow is as follows: The set of tasks are submitted to the scheduler, where the heuristic algorithm runtime is initiated in order to get the scheduled queue, i.e., which node for each task. The QoS controller runs the algorithm, and the scheduler then sends the schedule order to the cloudcore based on the queue. The selected edge node receives the schedule order for each task and then initiates the pod. We introduce Algorithm 1, a heuristic algorithm called STaSA (Service–Time-Aware Scheduling Algorithm for multi-node KubeEdge cluster) based on an enhanced version of the ACO.
**Algorithm 1.** STaSAInput: *p*; *Max_Loop*; *Ants_N*; Q; $\beta ,\;\gamma ,\;\rho ;\;\varphi_{1};$
$time_{j}$; $v_{mem}^{f,t}$, $v_{cpu}^{f,t}$
$e_{n}$.**Output:** Placement of pod *i* on node *j*1: Initialize parameters {Max_Loop,Ants_N, Q, α, β, γ,ρ}2: If $\left(v_{cpu}^{f,t}+v_{mem}^{f,t}>R\right)$ then   Instantiate new set of nodes using Equation (7)//include check capacity condition of nodes states 3: End if4: Initialize the pheromone trail using Equation (13)5: Initialize the placement cost using Equation (9) 6: For *nloop* from 1 to *Max_Loop* do7:   Random shuffle input pod queue8:    For *ant_k* from 1 to *Ants_N* do9:     For *i* from 1 to *p* do10:      Calculate the time value based on (12) 11:      Calculate the probability of placement of pod *i* on each node *j* using12:       Equation (13)       //ant_k chooses node *j* for pod *i* according to the highest probability      Add the selected node *j* to the schedule table as a placement of pod *i* for *ant_k*13:     End for14:     Calculate the cost of pod *i* for *ant_k* using Equation (10)15:    End for16:   Update the pheromone trail using Equation (15) 17:  End for18: Repeat until the maximum number of iterations is reached or best placement found

The introduced heuristic algorithm is based on an enhanced version of the ACO. STaSA starts by calculating the total resources available in the instantiated nodes. If the number of available resources is below the number of resources requested by the set of pods *p*, a new subset of nodes is instantiated then instantiated based on Equation (7) that prioritizes the node utilization. STaSA begins by calculating the initial pheromone value $\tau_{ij}$ and the placement cost matrix using Equations (9) and (12), respectively. With every loop and based on probability value Equation (11), each ant finds a placement for the set of pods *p* on nodes *N*. The maximum probability threshold method is applied for the selection of node *j*, where a random number ϵ [0,1] is generated and an aggregate probability value derived. The aggregate sum of probabilities is then arranged in ascending order, and the cumulative probability equal to or greater than the generated random number is selected. The pheromone trail is updated after each loop. Algorithm 1, terminates when either the maximum number of iterations is reached or by reaching the local minimum, i.e., the best placement.

## 8. Evaluation

We implemented our scheduler algorithm in the Python programing language. A cluster is set up, including a set of several nodes with KubeEdge version 1.4. The master node is configured with 8 CPU cores and 8 GB of RAM, and worker nodes, run with 4 CPU cores and 4 GB of RAM, more specifically deployed on Intel Xeon e5 2640, 2.6 GHz, 64 GB RAM, and 32 logical cores server. The test-bed configuration is shown below in Table 3. We evaluate the scheduling quality of STaSA in two ways: (i) compare with the previously implemented ACO baseline scheduler, FCFS algorithm (described in Section 3) using production workloads, and (ii) study critical evaluation metrics and use these to compare the quality. For evaluation, we use memory utilization, CPU utilization, service time, and cost metrics as comparison parameters. For comparison, we benchmark with ACO and FCFS algorithms. We estimate the cost of each approach based on the billing model of existing cloud providers [34]. We assume a per-second billing of USD 0.011 for each worker node based on Microsoft Azure’s general purpose B2S instance type, with any partial use being rounded up to the nearest second. The workload consists of tables that describe the submitted task requirements, machine configurations, and task resource usage. The scheduling table contains information such as time, pod ID, and task ID. Additionally, the table includes normalized data, such as resource requests for CPU cores and RAM. We conducted several trials as part of our experiment with the input values shown in Table 4.

## 9. Experimental Results

In our implementation environment, we investigate the three aspects of utilization, service time, and cost. Consequently, three algorithms have been compared, and the results are shown in Figure 4, Figure 5, Figure 6, Figure 7 and Figure 8. CPU utilization is investigated in Figure 6, where it is shown that STaSA outperforms the other two algorithms. As can be seen by the increase in the number of pod instances, the utilization value is slightly above average for all the algorithms but improves in STaSA as the ants are able to find the nearest optimal solution mapping for the pods to nodes. This indicates the minimum fitness value has been reached. ACO also shows good performance, but FCFS with imbalanced processing performs badly when computing slower tasks. Additionally, considering that all the workers operate in parallel, STaSA can keep the CPUs busy due to a fast upload resulting in higher resource utilization.

Memory utilizations among different nodes in a cluster system are highly unbalanced in practice where page faults, i.e., memory misses, might occur in some heavily loaded nodes. This is demonstrated well in Figure 5. When the number of requests is low, the utilization rate for all the algorithms is relatively lower. However, with increasing pod requests, STaSA is able to attain a higher utilization and slightly outperforms ACO. Their solutions converge at a point when the pods are between 30 and 35, but STaSA is able to minimize page faults and produces better results.

As can be seen from Figure 6, when the number of pods is low, the pod service of the three algorithms is relatively minimal. With the increase in the number of pod instances, the service time of STaSA and ACO is significantly better than FCFS. This is due to STaSA and ACO being able to converge faster and offer a better performance solution for the combinatorial optimization problem. Simultaneously, we observe that the service time of STaSA is slightly better than baseline ACO.

Figure 7 shows the total costs of the workload for the three algorithms. We can see that when the difference in cost between STaSA and ACO is relatively small compared to FCFS. This observation can be explained from the perspective of the relationship between cost and utilization factors. Additionally, our approach adjusts the corresponding pheromone value that is considered in the objective function. This results in a better solution with the cost as a constraint. The service time of the three algorithms is used as a basis to observe the QoS performance. We monitor the algorithms under the same QoS requirement to determine their efficiency. A function instance is considered unavailable or busy while processing a request, and if there are no other available instances of that function, then the scheduler is unable to schedule and run another request for the same function until an instance becomes available. Those are considered unscheduled pods. We can observe in Figure 8 that STaSA outperforms the other two algorithms throughout. It demonstrates very low violations, which means that it has a higher success rate compared to the other two.

## 10. Conclusions

In this study, we proposed a dynamic pod scheduling model to solve the task scheduling problem at the edge. This approach considered user QoS requirements as a primary goal in the solution. Containers were adopted to provide the computational resources to the application requests. Based on application and container characteristics, we modeled the pod scheduling process and proposed a novel pod scheduling algorithm, STaSA, that is based on a modified ant colony optimization model. The multi-objective goal seeks to maximize node utilization, minimize the cost, and optimize the service time. The constraints included resource capacity (CPU and memory) and total service time. Experiments conducted showed that STaSA outperformed two other methods, namely ACO and FCFS. However, to simplify the service model, we ignored the network transmission, which should be considered in a larger deployment scenario setup. Furthermore, task dependencies and application partitioning for efficient computation are not considered as identifying the resource-intensive components in workloads is a complex issue.

## Figures and Tables

**Figure 1 sensors-21-05797-f001:**
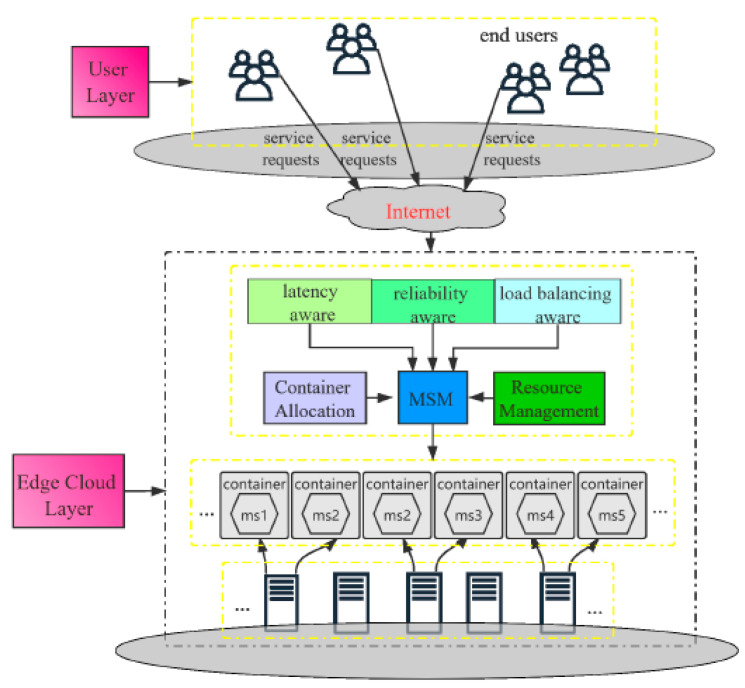
Containerization deployment framework.

**Figure 2 sensors-21-05797-f002:**
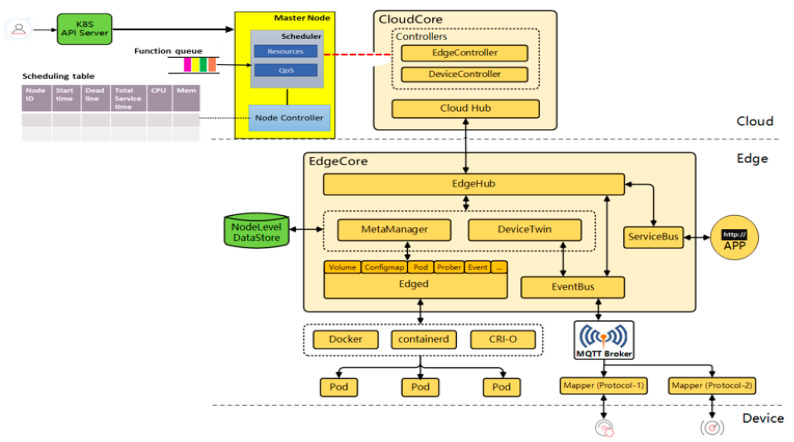
Implementation in KubeEdge platform.

**Figure 3 sensors-21-05797-f003:**
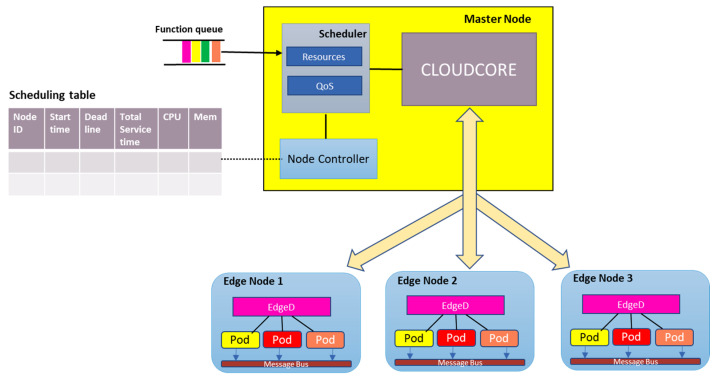
High-level scheduler architecture.

**Figure 4 sensors-21-05797-f004:**
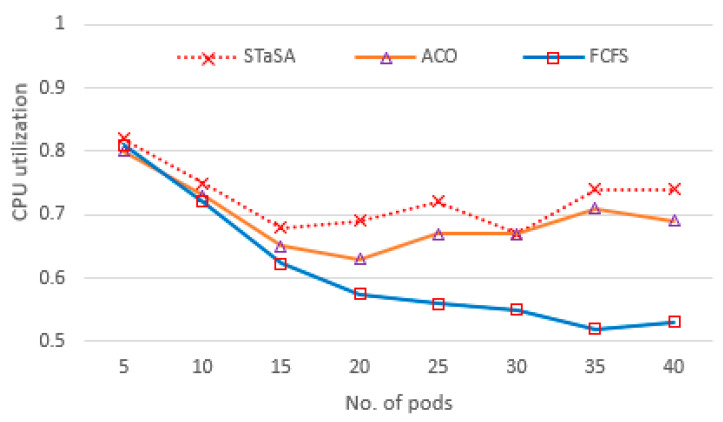
CPU utilization.

**Figure 5 sensors-21-05797-f005:**
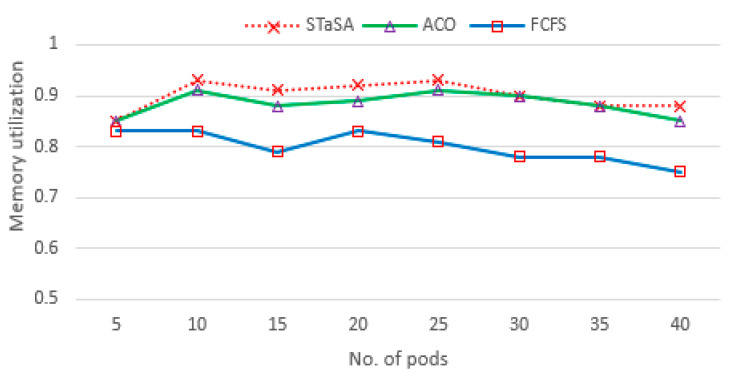
Memory utilization.

**Figure 6 sensors-21-05797-f006:**
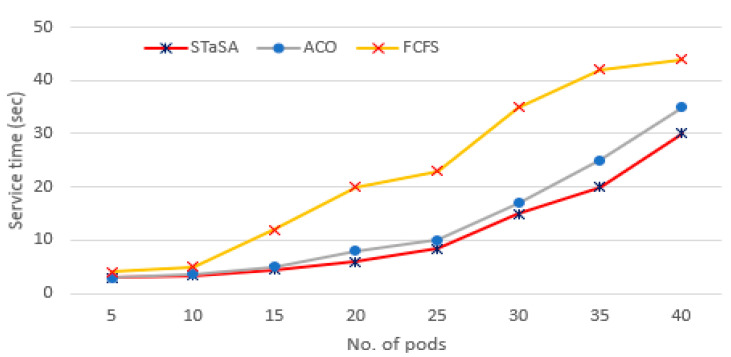
Pod service time.

**Figure 7 sensors-21-05797-f007:**
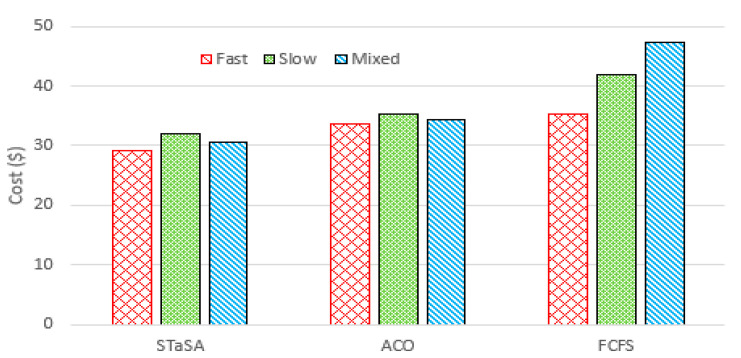
Cost for different workloads.

**Figure 8 sensors-21-05797-f008:**
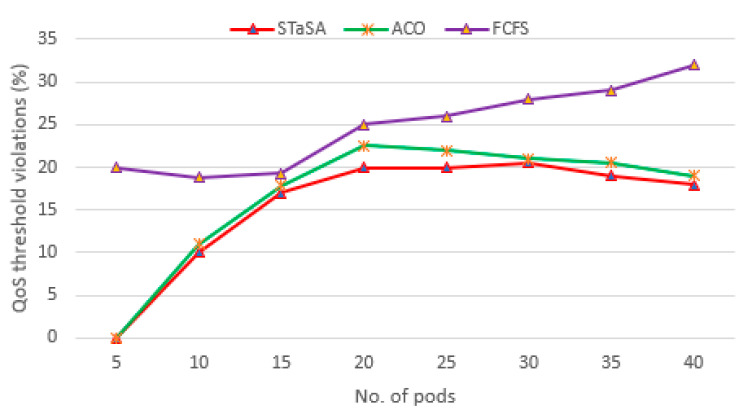
A comparison of QoS violations.

**Table 1 sensors-21-05797-t001:** Parameter notations.

Parameters	Description
*N*	Set of nodes j=1,2,…, n | n ∈N
*P*	Set of pods i=1,2,…, p | p ∈P
*F*	Set of functions
*R*	Set of resources
$r_{cpu}^{j,t}$	Total CPU capacity of node *j* at time *t*
$r_{mem}^{j,t}$	Total memory capacity of node *j* at time *t*
$v_{cpu}^{f,t}$	CPU requirements of function *f* at time *t*
$v_{mem}^{f,t}$	Memory requirements of function *f* at time *t*
${\hat{r}}_{cpu}^{j,t}$	CPU capacity of node *j* at time *t*
${\hat{r}}_{mem}^{j,t}$	Memory capacity of node *j* at time *t*
$\delta_{cpu}$ $,\;\delta_{mem}$	Default CPU and memory capacities, respectively
$x_{f,i,t}$	Binary variable for instantiation of function *f* as pod *i* at time *t*
$y_{ij}$	Binary variable associating pod *i* to node *j*
$U_{j}^{t}$	The utilization of node *j* at time *t*
$C_{node}$	Deployment cost of any pod *i* on any node *j*
$time_{j}$	The processing time of a pod *i* on node *j*
$D_{if}$	Critical deadline for executing a pod *i* of function *f*
*L*	Objective function

**Table 2 sensors-21-05797-t002:** Comparison summary of ACO-related research studies.

Ref	Objective	Heuristic Used	Comparison Algorithm	Performance Metrics	Difference	Platform	Application Environment
[31]	Improve resource utilization in terms of CPU cores and memory for VMs and PMs Minimize number of instantiated VMs and PMs	ACO-BF	Max-fit Best-fit	Memory utilization CPU utilization	Service time not considered Cost not considered	Docker	Cloud
[28]	Maximize application performance	ACO	Greedy	Resource reservation Workload performance	Cost not considered Few containers deployed	Docker	Edge
[30]	Reduce the network transmission overhead among microservices Load balancing in the physical nodes	ACO_ MCMS	GA_MOCA Multiopt	Cluster Resource load Network transmission overhead	Consider bandwidth overhead	Docker	Cloud
[29]	Improve response time and throughput	MACO	FCFS	CPU utilization Energy consumption	Consider energy consumption QoS not factored	Docker	Cloud
Our approach	Improve resource utilization in terms of CPU and memory Minimize the response time for scheduled pods Minimize cost of pod placement	STaSA	ACO FCFS	Resource utilization Percentage QoS violations Response time	QoS considered Cost considered Service time considered	KubeEdge	Edge

**Table 3 sensors-21-05797-t003:** Test-bed settings.

Entry	Configuration
Physical Servers (4)	Master Node (1): RAM: 8 GB; CPU: 8 cores Worker Nodes (3): RAM: 4 GB; CPU: 4 cores
Container OS	KubeEdge

**Table 4 sensors-21-05797-t004:** Input values.

Parameters	Description	Value
*Pods*	Number of pods	40
*Ants_N*	Number of ants	8
*Q*	Heuristic constant	1
*α*	Heuristic constant	0.1
*β*	Heuristic constant	2
*γ*	Heuristic constant	0.3
ρ	Pheromone evaporation rate	0.2
$\varphi_{1}$	Weight factor	0.5
$v_{cpu}^{f,t}$	CPU requirements of function f	[100–1000] ms
$v_{mem}^{f,t}$	Memory requirements of function f	[100–1000] MB
$e_{n}$	Node price per second	$0.11

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
