# Peer review of "QoS-Based Service-Time Scheduling in the IoT-Edge Cloud"

_sensors, 2021, doi:10.3390/s21175797_

Round 1

Reviewer 1 Report

  1. Combing too many known techniques can not be considered as novelty, unless the method is justified to be either (i) optimal, or (ii) if not optimal, but it shows a huge performance over the previously proposed techniques. Finally, if the paper implements an existing method, then the experiments should be comprehensive and exhaustive. The manuscript is somehow short from this perspective.
  2. Clarify what you mean by heterogeneity and task independency, then explain how you have factored in these concepts in your model.
  3. Scheduling and optimizing the overheads in similar problems have been addressed extensively in the literature. Clarify what contributions you have in this fold that makes your paper novel.
  4. QoS has different interpretations in different systems. I would suggest being more specific when you define your contributions, including in the title and also in the abstract.
  5. In the bullet items in the contributions subsection (page 2), I was not able to recognize a significant novelty in the first two items. I would suggest rewriting them and being more specific about your contributions. If “deployment” of the previously proposed idea and also using the open-source platform, i.e., KubeEdge, is among your main contributions, then it should be mentioned in the contributions.
  6. Explain better/justify how ACO can optimally optimizes your problem.
  7. Also, the probability in Eq. (12) deserves a better explanation (or need to be cited). Since the probability changes in each iteration, it should be clarified how you update this value in your algorithm.
  8. The relationship with EDF is not clear in Algo. 1. It looks like a very general pseudocode to me in the algo. How do you factor in this command?
  9. Algorithm 1 (STaSA) needs more commenting that explains the process better.
  10. How do you define max_loop in Algo. 1?
  11. Although an ACO will converge at some point, but how do you ensure the timely convergence based on your time sensitive constraints?

Reviewer 2 Report

The paper presents an enhancement of ant colony optimization (ACO) for the problem of QoS based scheduling of services in the IoT-Edge cloud; a series of experiments show significant improvements over a base-line ACO algorithm and a first-come first-serve algorithm.

On the positive side, the experimental evaluation is clear and convincingly demonstrates the effectiveness of the approach; thus the results basically deserve publication. On the negative side, however, there are some issues that do not allow me to recommend publication of the paper in its current form:

  1. (Major) The mathematical presentation given in Section 3 (and at the beginning of Section 4) is poor and contains many errors (more details below); the quality is way beyond the threshold of a journal presentation. The mathematical material must be considerably revised.
  2. (Medium) The presentation of the algorithm in Section 4 is sketchy and does not clearly differentiate the enhancements of the presented algorithm from the base-line ACO algorithm. In particular, the presentation of Algorithm 1 should be made more concrete by referring to the variables introduced in Section 3 and the last paragraphs of Section 4 should be revised to highlight the enhancements of the algorithm to the base-line algorithm.
  3. (Minor) Section 3 contains a subsection "Scheduling Model" which better belongs to Section 2 "Related Work" ("Works" -> "Work") as does most of the material of Section 4 before the subsection "STaTA Scheduler".

Deficiencies in Section 3:

  • line 171: "a cluster nodes N with resources R. Users may requests" -> "a cluster of  n nodes with p pods. Users may request" (the values N and R do not play any role in the further formalization, but the number p of pods does)
  • line 175: "x_f,p,t" -> "x_f,i,t"
  • line 177: "phi_p^t" -> "phi^t" (the quantity does not depend on p)
  • equation (1): "sum_F sum_P sum_T" -> "sum_{f in F} sum_{p in P} sum_{t in T}"
  • equation (2): "sum_{n in N,p in P}" -> "sum_{i=1}^p sum_{j=1}^n"
  • equations (3,4): "r_cpu^{j,t}" and "r_mem^{j,t}" these quantities should be explained in the text, referring to both j and t in the explanation (also the corresponding lines in Table 1 should be updated to refer to j and t)
  • equations (5,6): "sum_{j=1}^n" -> "sum_{i=1}^p"
  • line 187: "We define utilization" -> "We define the utilization U_j^t"
  • equation (11): "L" -> "L_j^t". This quantity should be explained in the text and Table 1, referring to both j and t.
  • equation (11): "time_j". This quantity is nowhere defined or explained (actually it will be introduced later in Section 4). Should be described in the text and Table 1.
  • Table 1: "p_{i,j}^{f,t}" nowhere used, can be dropped.

Section 4:

  • As mentioned above, the first three pages belong to "Related Work". The description "It is a probability-based approach ... In the search for food, an ant ..." This description lacks context. First introduce the topic of "ant colony optimization (ACO)".
  • line 242. "Where ..." -> "Here ..."
  • line 242. "tau_{i,j}^t" -> "tau_{j}^t" (the quantity does not depend on i)
  • line 243. "time_j" is introduced here but already used in Section 3.
  • equation (12): "p_{i,j}^k" -> "p_{i,j}" (the quantity does not depend on k),
  • equation (12): why the conditional definition? generally j only runs over values 1..n, why not here? I suggest to define it unconditionally.
  • equation (12): alpha, beta, gamma not explained.
  • equation (13): "tau_{i,j}(t)=U_j^t" I suspect "tau_{i,j}(0)=U_j^0" is meant, since otherwise the recurrence (14) lacks a base case.
  • equation (14): "tau_{i,j}^k(t+1)" -> "tau_{i,j}(t+1)" (the quantity does not depend on k)
  • equation (14): "sum_{k=1}^8" -> "sum_{k=1}^A" where A is the number of ants (should be explained in the text)
  • equation (14): "delta tau_{i,j}^k(t)" I strongly suspect something is wrong here, since this quantity is defined as a constant 1/L, thus the sum would be just a fixed multiple of this constant. I guess it should read as "delta * tau_{i,j}(t)" and equation (15) should be "delta := 1/L".
  • line 296: "equation (time equation)" adequate reference missing.
  • line 293: "Pods" -> "p" (number of pods)
  • line 311: drop this, parameters are assumed to be initialized by the caller.
  • line 321: "P" -> "p"
  • line 322: "based on EDF" unclear what this means.
  • lines 324 and 325: wrong indentation
  • Generally Algorithm 1 is sketchy and should be refined to refer to the variables introduced in Section 3.

Section 5:

  • line 346: "Python environment" -> "Python programming language", "setup" -> "set up".
  • line 348 and Table 3: "CPU cores", please give the type of the cores and the clock rate.
  • Table 4: "millicpu" -> "ms", "mb" -> "MB"
  • Figures 4,5,6,8: blurry, please use higher resolution.

Round 2

Reviewer 1 Report

My comments have been addressed. 

Reviewer 2 Report

My previous comments have been mostly adequately addressed (the algorithm is still a bit sketchy). Some minor corrections are given below.

  • Line 190: "users may requests" -> "request"
  • Equation (1): "p in P"-> "i in P"
  • Table 1: "C_node" "deployment cost of pod i on node j". This cost is apparently the same for every node i and j (C_node is a constant), thus better write "of any pod i on any node j"
  • Equation (17): drop "(t)", not used here or in eq. (16). A comment should be made why the defined quantity does not depend on i,j,k such that in fact eq. (16) defines simply the quantity A*(Q/L).
